# Investigation into Freezing Point Depression in Soil Caused by NaCl Solution

**Feng Ming [1], Lei Chen [1,2,*], Dongqing Li [1] and Chengcheng Du [3]**

1   State Key Laboratory of Frozen Soil Engineering, Northwest Institute of Eco-Environment and Resources, Chinese Academy of Sciences, Lanzhou 730000, China; mingfeng05@lzb.ac.cn (F.M.); dqli@lzb.ac.cn (D.L.)

2   College of Resources and Environment, University of Chinese Academy of Sciences, Beijing 100049, China

3   Gansu Province Transportation Planning, Survey and Design Institute Co. Ltd., Lanzhou 730030, China; 13669360262@163.com

*   Correspondence: chenlei8@lzb.ac.cn; Tel.: +86-93-1496-7947

**Abstract:** Engineering practices illustrate that the water phase change in soil causes severe damage to roads, canals, airport runways and other buildings. The freezing point is an important indicator to judge whether the soil is frozen or not. Up to now, the influence of salt on the freezing point is still not well described. To resolve this problem, a series of freezing point tests for saline soil were conducted in the laboratory. Based on the relationship between the freezing point and the water activity, a thermodynamic model considering the excess Gibbs energy was proposed for predicting the freezing point of saline soil by inducing the UNIQUAC (universal quasi-chemical) model. The experimental results show that the initial water content has little influence on the freezing point if the initial water content is higher than the critical water content, while the freezing point decreases with the decrease of the water content if the initial water content is lower than the critical water content. Moreover, it is found that the freezing point is related to the energy status of liquid water in saline soils and it decreases with the increase of the salt concentration. Moreover, the freezing point depression of saline soil is mainly caused by the decrease of water activity. Compared with the other two terms, the residual term, accounting for the molecular interactions, has an obvious influence on the water activity. This result is helpful for understanding how salt concentration affects the freezing point of saline soil and provides a reference for engineering construction in saline soil areas.

**Keywords:** saline soil; freezing point; chemical potential; water activity; UNIQUAC model

---

## 1. Introduction

Saline soil is a general term of saline soil and alkaline soil. The statistic results show that the saline soils are widely distributed throughout the world, covering the tropical and cold regions. In China, the saline soils are mainly distributed in the arid and semi-arid areas. Meanwhile, in these areas, also distributed is a large area of permafrost and seasonal frozen soil [1]. The engineering practices demonstrate that soil deformation induced by the salt expansion and the water phase change leads to severe damage to roads, canals, airport runways and other buildings in these arid and semi-arid areas [2,3].

The freezing point has many practical uses. In food engineering, the freezing point is a key parameter to confirm whether the food is in a low temperature storage [4]. In the refrigeration equipment, the solutes are added in the radiator fluid to reduce its freezing point to make sure that the radiator does not freeze at a cold environment [5]. In the geotechnical engineering, the freezing point of soil is important for engineering construction and applications of artificial freezing technical [6], owing to the freezing point being the key point to judge whether the soil is frozen or not. In addition,

in cold regions, the road de-icing technology takes advantage of this depression effect to lower the freezing point of ice [7]. As seen, an accurate freezing point of soil is important parameter for practices.

It is known that the freezing point of pure water is 0 °C. However, the measured freezing point of soil is always below 0 °C. This is mainly caused by the soil particle surface free energy and the existence of chemical substances. This is why the measurement of the freezing point becomes difficult [8]. Since the Workman–Reynld effect was proposed in the 1950s, the frozen potential of solution was gradually introduced in the research of frozen soil. However, it was found that there was a sudden potential change at a temperature that was not the freezing point. Therefore, the freezing point determined by the sudden potential change was incorrect. As an improving test method, the thermistor was used to measure the freezing point [9]. Through this method, the freezing process and freezing point at different conditions can be determined [10]. The results show that the freezing point almost increased linearly with the increasing water content, which was consistent with the results found by other researchers [5]. After that, the Differential Scanning Calorimetry method (DSC) was recently used to measure the freezing point of frozen soil. In conclusion, the freezing point of saline soil can be measured in the laboratory. However, due to the diversity of solution composition, the experimental task is huge. Therefore, it is necessary to study the freezing point from the aspect of theory.

It is found that the addition of salt can effectively reduce the frost heave of soil [11]. This phenomenon has been simulated in the laboratory tests. It was found that the presence of salt resulted in the water to be frozen at a lower temperature. This indicated that the freezing point was decreased by salt in soil. Furthermore, the influence of different salt types on the freezing point was different [12]. Later on, a systematic studies on the freezing point of saline soil were conducted in the laboratory [13]. The results indicated that the freezing point decreased with the increase of salt content and with the decrease of water content. For a given water and salt content, the freezing point of fine particle soil was lower than that of coarse particle soil. For the soils with a lower salt content, the unfrozen water content of soils increased with the increase of salt content [14]. Through the laboratory experiments, the change law of the freezing point under the effect of salt can be easily obtained and discussed [13,14]. However, scarce literature exists on how salt affects the freezing point of soils.

The aim of this study was to investigate how salt reduces the freezing point of soil. To achieve this goal, this paper was organized as follows. Firstly, we introduce the concepts related to the freezing point of soil, and the methods used to measure the freezing point. Secondly, based on the experimental results, the change law of freezing point was analyzed. Thirdly, the UNIQUAC model was employed to calculate water activity in soil solution and a general formula for predicting the freezing point of the saline soil was proposed. Finally, the influence of the three components (in the excess Gibbs energy) on water activity was discussed. This result of this research is helpful for understanding how the salt affect the freezing point of soil and other porous media.

## 2. Materials and Methods

### 2.1. Experimental Material

The soil used in this study was collected from the Loess Plateau in Lanzhou (36°03′ N, 103°40′ E). To investigate the effect of salt content on the freezing point, all the soil was washed by distilled water to eliminate the salt. The desalinated soil was dried at 105 °C for 24 h and sieved through a 2-mm sieve. The soil particles with sizes of less than 2 mm were used as the experimental specimens. The particle size distribution curve was shown in Figure 1. The silty clay has a plastic limit of 15.4%, a liquid limit of 28.7% and a specific gravity of 2.62 g/cm$^3$. According to the WRB-based soil classification [15], the soil can be classified into the aridic.

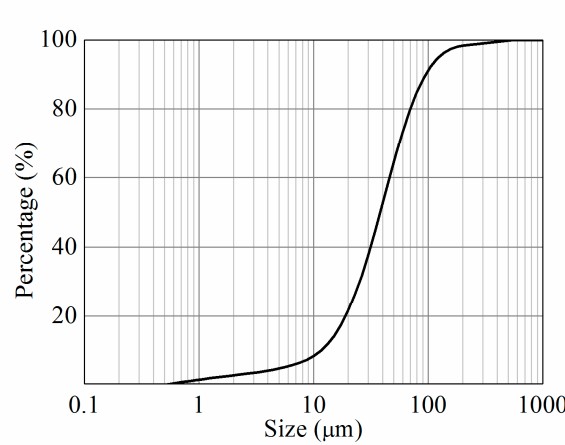

**Figure 1.** Particle size distribution curve of the tested soil.

## 2.2. Sample Preparation

Based on the results of the field survey, dry density, water content and salt concentration of the specimen were determined as 1.0~1.5 g/cm$^3$, 5~50% and 0~4%, respectively. According to the test scheme, the specimen was mixed with different mass of water, salt (NaCl) and soil (without water) at room temperature and sealed for 24 h to make the sample reach a uniform state.

## 2.3. Experimental Scheme

The freezing point of saline soil was measured by a self-made device (made by the State Key Laboratory of Frozen Soil Engineering, Lanzhou, China). The schematic of the test apparatus for freezing point measurement was presented in Figure 2. As seen, the test apparatus mainly consists of sample box, sample cover, temperature sensor (with an accuracy of ±0.02 °C) and data acquisition (datataker 80, Thermo Fisher, Canberra, Australia). The cold bath (with an accuracy of ±0.1 °C, type: FP55-SL, made by Julabo, Allentown, PA, USA) was used to provide a sub-freezing temperature. The sample box was 5 cm in high and with an inner diameter of 3.5 cm. The sample box and the cover were made of cooper. In order to inset the temperature sensor, there was a hole in the sample cover. The gap between the temperature sensor and the sample cover was filled with rubber.

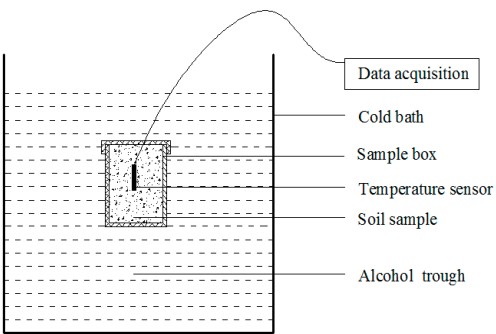

**Figure 2.** Test apparatus for freezing point measurement.

After the specimen was put into the sample box, the sample cover was covered on the sample box. Then, a temperature sensor was inserted into the sample box. The gap between the temperature sensor and the sample cover was filled with rubber. After that, the sample box was placed in the cold bath at 15 °C for twelve hours. Finally, the temperature of the cold bath was set as −15 °C. The temperature data were recorded automatically by the data acquisition system with an interval of 10 s.

*2.4. Determination of Freezing Point*

The crucial determinant of water phase change was freezing point. As seen in Figure 3, the cooling curve can be divided into four stages according to the temperature change rate. When the temperature of water reached a certain supercooling value, the crystallization process was started. The temperature of starting crystallization was called the actual crystallization temperature. With the crystallization ongoing, the latent heat was released and resulted in the soil temperature raised up. If the released latent heat is enough to cause the soil temperature rise instantaneously, there will be an obviously temperature jump. If not, there is no temperature abrupt transition stage. With more nucleus formation, the pore water and pore ice achieved an equilibrium state. The temperature at this equilibrium state almost kept constant and was taken as the freezing point.

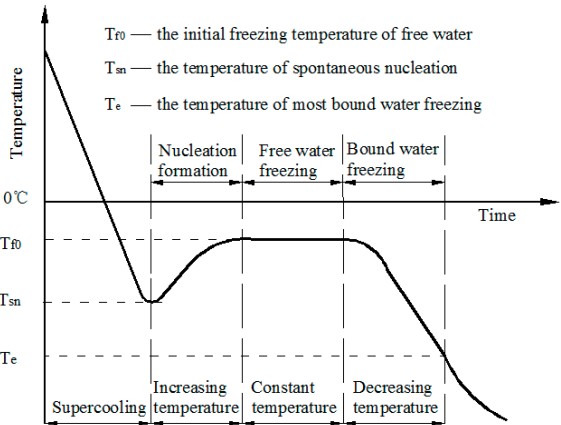

**Figure 3.** Cooling curve for the soil–water system.

## 3. Results

Based on the experimental results, the freezing point can be obtained at different test conditions. Figure 4 shows the variation of soil temperature with the different salt concentrations. As shown, compared with the higher concentration, there is a clear temperature rise when the concentration is in a lower state. With the increase of salt concentration, water is hard to crystallize. As a result, the released latent heat is small, and the soil temperature rises a little. In addition, the time point of the abrupt transition of temperature is postponed with the increasing salt concentration.

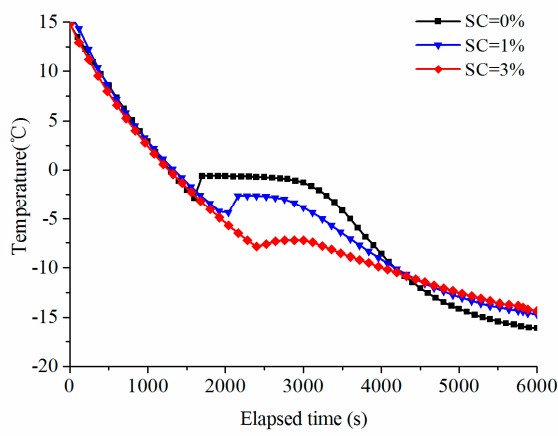

**Figure 4.** Cooling curves of saline soil with different salt concentrations (SC: salt concentration).

Figure 5a shows the influence of water content on the freezing point. As seen, when the water content is lower than the critical water content, the freezing point increases with the increase of water

content. When the water content is larger than the critical water content, the water content has little influence on the freezing point. Generally, at a lower water content, the smaller capillary tubes were filled with water. According to the capillary theory, the smaller capillary radius has a lower freezing point. With the increase of water content, the capillary tubes with larger size were filled with water. The Gibbs–Thomson equation indicated that the larger capillary radius had a higher freezing point. When the water content exceeds the critical water content, the water content has little influence on the size of capillary tubes. Consequently, the freezing point has little change.

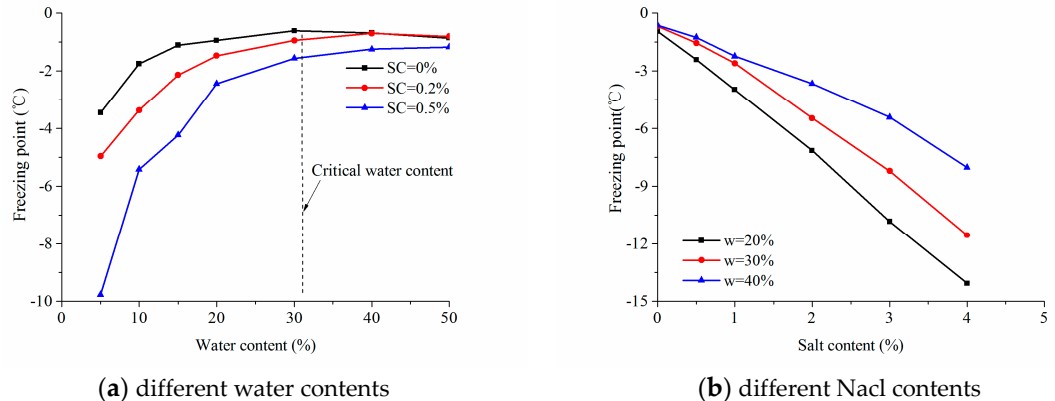

(**a**) different water contents  (**b**) different Nacl contents

**Figure 5.** Influence of water content and Nacl content on the freeing temperature.

Figure 5b presents the influence of NaCl concentration on the freezing point. Generally, the freezing point decreases with the increase of salt concentration. At the same salt concentration, the larger water content has a higher freezing point. Moreover, if the NaCl solution is treated as an ideal dilute solution, then the relationship between the freezing point and the salt concentration can be expressed in a simple form [10],

$$\Delta T_F = T_{F0} - T_F = k_r b_r \beta, \tag{1}$$

where $T_{F0}$ is the freezing point of pure water; $T_F$ is the freezing point of solution; $k_r$ is the cryoscopic constant, $k_r = 1.853 \,°C/(mol \cdot kg^{-1})$ for water; $b_r$ is the mass molarity of solute, $mol \cdot kg^{-1}$; $\beta$ is the van't Hoff factor ($\beta = 2$ for NaCl).

For the NaCl solution, there is $1 \, mol \cdot kg^{-1} = 1 \, mol \cdot L^{-1}$. As a result, the decrease ratio of freezing point $\Delta T_F$ with the concentration $b_r$ can be calculated by Equation (2),

$$\Delta T_F = 1.853 \times 2 = 3.706 \,°C/(mol \cdot L^{-1}), \tag{2}$$

Redrawing Figure 5, the relationship between $\Delta T_F$ and salt concentration can be obtained. It can be seen from Figure 6 that the freezing point of saline soil decreases linearly with the increase of NaCl concentration with a slope about $-3.866 \,°C/(mol \cdot kg^{-1})$. This slope $-3.866 \,°C/(mol \cdot kg^{-1})$ is close to that of the ideal dilute solution $-3.706 \,°C/(mol \cdot kg^{-1})$. Therefore, if the freezing point of desalinated soil is known, then the freezing point of saline soil can be calculated by Equation (1).

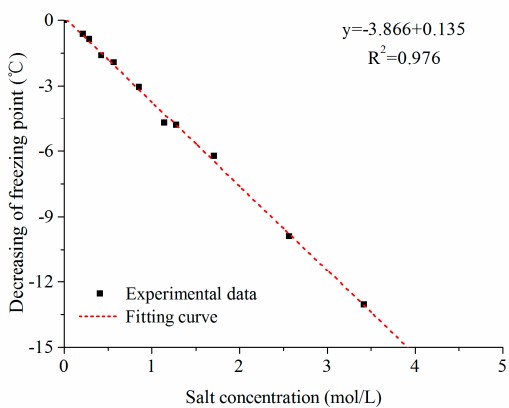

**Figure 6.** Influence of NaCl concentration on freezing point depression.

## 4. Theoretical Model

### 4.1. Thermodynamic Model

The experimental results indicate that the freezing point of soil can be decreased by adding solute (such as, NaCl and $Na_2SO_4$) and the decreasing value can be calculated by the empirical formula Equation (1). However, this simple relation cannot reflect the influence mechanism of the solute on the freezing point. In addition, how the salt affects the freezing point is not clear. To resolve this problem, the theoretical basis around the freezing points was investigated in this section.

According to the previous study [16], the chemical potential of the liquid and solid solvent can be expressed as,

$$\mu^w(T,P) = \mu_0^w(T,P) + RT \ln a_w(T,P,\phi_w),\tag{3}$$

$$\mu^i(T,P) = \mu_0^i(T,P) + RT \ln a_i(T,P,\phi_i),\tag{4}$$

where $\mu^w(T,P)$ and $\mu^i(T,P)$ represent the chemical potential of pore water, pore ice at the state $(T,P)$, respectively; $\mu_0^w$ and $\mu_0^i$ represent the chemical potential of pore water, pore ice at a reference state, respectively; $R$ is gas constant; $T$ is temperature; $P$ is the stress; $a_w$ is the water activity; $\phi_w$ and $\phi_i$ represent the parameters of pore water and pore ice, respectively. At the equilibrium state, the chemical potential of pore water must be the same in liquid phase and solid phase, then the following formula can be written,

$$\mu_0^i = \mu_0^w,\tag{5}$$

At the same temperature, the difference between the chemical potential of pore water and pore ice can be calculated by Gibbs energy of fusion $\Delta G_{fus}^0$. Combined with Equations (3)–(5), $\Delta G_{fus}^0$ can be expressed as,

$$\Delta G_{fus}^0 = \mu_0^w - \mu_0^i = -RT \ln a_w,\tag{6}$$

According to the Gibbs–Helmholtz equation, the relationship between the chemical potential and the temperature can be described as,

$$\left(\frac{\partial G/T}{\partial T}\right)_P = -\frac{H}{T^2},\tag{7}$$

Combining Equations (6) and (7), there is,

$$\frac{\mathrm{d}(\ln a_w)}{\mathrm{d}T} = \frac{\Delta H_{fus}}{RT^2},\tag{8}$$

where $\Delta H_{fus}$ is the enthalpy change upon fusion at the lower freezing point. According to previous study [17], $\Delta H_{fus}$ can be expressed as,

$$\Delta H_{fus}^{T_F} = \Delta H_{fus}^{T_{F0}} + \Delta C_{fus}(T_F - T_{F0}), \tag{9}$$

$$\Delta C_p = C_{p,m}(\text{liq}) - C_{p,m}(\text{solid}), \tag{10}$$

where $C_{p,\phi_w}$ and $C_{p,\phi_i}$ are the specific heat capacity in the liquid and solid state, respectively. The freezing point of pure water is around the temperature $T_{F0}$ ($a_w = 1$). The freezing point of the solution is $T_F$. When the water activity is $a_w$, the integration of Equation (8) is

$$\int_1^{a_w} \frac{d(\ln a_w)}{dT} = \int_{T_{F0}}^{T_F} \frac{\Delta H_{fus}^{T_F}}{RT^2}, \tag{11}$$

Substituting Equations (9) and (10) into Equation (11), and rearrange Equation (11),

$$R \ln a_w = \Delta H_{fus}^{T_{F0}}\left(\frac{1}{T_{F0}} - \frac{1}{T_F}\right) + \Delta C_{fus}\left[\ln\left(\frac{T_{F0}}{T_F}\right) - \frac{T_{F0} - T_F}{T_F}\right], \tag{12}$$

To eliminate the logarithmic function, the Taylor series expansion (Equation (13)) was introduced,

$$\ln(1 + y) = y - y^2/2 + y^3/3 + \cdots, \tag{13}$$

where $y$ is variable. It can be found that the third term and the terms after the third one in the right part of Equation (13) have a little influence on the whole value. As a kind of approximately treated, we just reserve the first two terms of Equation (13). Substituting the first two terms of Equation (13) into Equation (12). There is,

$$R \ln a_w = \Delta H_{fus}^{T_{F0}}\left(\frac{1}{T_{F0}} - \frac{1}{T_F}\right) - \Delta C_{fus}\frac{(T_{F0} - T_F)^2}{2T_F^2}, \tag{14}$$

By introducing the concept of freezing point decrease $\Delta T_F$,

$$\Delta T_F = T_F - T_{F0}, \tag{15}$$

Substituting Equation (15) into Equation (14), there is,

$$R \ln a_w = \Delta H_{fus}^{T_{F0}}\left(\frac{1}{T_{F0}} - \frac{1}{T_{F0}\Delta T_F}\right) - \Delta C_{fus}\frac{\Delta T_F^2}{2(T_{F0} + \Delta T_F)^2}, \tag{16}$$

Finally, $\Delta T_F$ can be calculated by Equation (17),

$$\Delta T_F = \frac{\Delta H_{fus}^{T_{F0}} - 2T_{F0} \cdot R \ln a_w - \sqrt{\left(\Delta H_{fus}^{T_{F0}}\right)^2 + 2\Delta C_{fus} \cdot R \ln a_w \cdot (T_{F0})^2}}{2R \ln a_w - 2\Delta H_{fus}^{T_{F0}}/T_{F0} - \Delta C_{fus}}, \tag{17}$$

For the pure water, the freezing point is $T_{F0} = 273.15$ K. Based on Equation (14), with the parameters present in Table 1, the relationship between the temperature and water activity can be obtained, as shown in Figure 7. As shown, when the water temperature $T = T_{F0} = 273.15$ K, the water activity is equal to 1. This indicates that the ice and water are in a coexist state. If water temperature $T > T_{F0} = 273.15$ K, then $a_w > 1$, it means the ice will thaw gradually into water. Otherwise, the water will freeze.

**Table 1.** Thermodynamic properties of pure water and methanol at 1 atm [18].

| Parameters and State | Value |
|---|---|
| Specific heat capacity, $C_{p,\phi_i}(T = 273.15)$ K, ice | 37.98  J/(K·mol) |
| Specific heat capacity, $C_{p,\phi_w}(T = 273.15)$ K, water | 75.92  J/(K·mol) |
| Enthalpy of fusion, $\Delta H_{fus}^{T_{F0}}(T = 273.15)$ K, solid water | 6010 J/mol |

**Figure 7.** Relationship between water activity and temperature.

*4.2. UNIQUAC Model*

The activity coefficient is an important parameter in the electrolyte solution theory and can be calculated by the UNIQUAC model. This model takes the excess Gibbs energy of the mixture as a function of the compositions [19]. A modified version of the UNIQUAC model was presented by Wisniewska and Malanowski [20]. This modified model can be used to predict the formation of mineral scale in geothermal and oilfield operations [21]. Details about the UNIQUAC model can be found in the original papers [19]. Here, we only introduce same necessary results.

In the UNIQUAC model, it is assuming that the excess Gibbs energy ($G_E$) consists of three parts: combinatorial ($G_C$), residual ($G_R$) and Debye–Hückel ($G_{DH}$).

$$G_E = G_C + G_R + G_{DH}, \tag{18}$$

Generally, the three parts of the excess Gibbs energy can be described by the following expressions. (1) Combinatorial term

$$G_C = RT \ \left[\sum_{i=1}^{n} \ln(\frac{\phi_i}{x_i}) - \frac{s}{2}\sum_{i=1}^{n} q_i x_i \ln(\frac{\phi_i}{\theta_i})\right], \tag{19}$$

where $n$ is total mole number; $x$ is mole fraction considering the species dissociated, mol/mol; $q$ is surface area parameter; $\phi$ is volume fraction can be calculated by Equation (20); $\theta$ is surface area fraction can be calculated by Equation (21); s is a coordination number and s $=$ 10. Subscripts $i, j$ are species $i, j$.

$$\phi_i = \frac{r_i x_i}{\sum_{j=1}^{n} r_j x_j}, \tag{20}$$

$$\theta_i = \frac{q_i x_i}{\sum_{j=1}^{n} q_j x_j}, \tag{21}$$

(2) Residual term

$$G_R = -RT \sum_{i=1}^{n} \ln \left( \sum_{j=1}^{n} \theta_j \tau_{ji} \right),$$ (22)

where $\tau$ and $u$ are parameters in UNIQUAC model defined as Equations (23) and (24), respectively.

$$\tau_{ji} = \exp \left[ \frac{-(u_{ji} - u_{ii})}{T} \right],$$ (23)

$$u_{ji} = u_{ji}^0 + u_{ji}^t (T - 298.15),$$ (24)

(3) Debye–Hückel term

$$G_{DH} = -RT \; x_s \, M_s \frac{4A}{b^3} [\ln(1 + bI^{\frac{1}{2}}) - bI^{\frac{1}{2}} + \frac{1}{2}b^2 I],$$ (25)

where $M$ is molecular weight, kg/mol; $b$ is constant parameter, 47.4342 kg$^{1/2}$/mol$^{1/2}$; $A$ is Debye–Hückel parameter presented by Nicolaisen et al. which is described as Equation (26) [22]; $I$ is ionic strength defined as Equation (27),

$$A = 35.765 + 4.222 \times 10^{-2}(T - 273.15) + 3.681 \times 10^{-4}(T - 273.15)^2,$$ (26)

$$I = \frac{1}{2} \sum_i m_i z_i^2,$$ (27)

where $m$ is molality, kg/kmol; $z$ is charge of ion $i$.

Considering the physical state of the pure specie and their standard state, the activity coefficients are related by Peralta et al. [17],

$$\ln \gamma_i^{ids} = \ln \gamma_i^{ps} - \ln \gamma_i^{\infty} i,$$ (28)

$$\ln \gamma_i^{\infty} = \lim_{x_s \to 1} (\ln \gamma_i^{ps}),$$ (29)

where $\gamma_i^{ids}$ and $\ln \gamma_i^{ps}$ are the activity coefficients of the species when ideal dilute solution and perfect solution are at the standard states, respectively. $\ln \gamma_i^{\infty}$ is the activity coefficient of the species at infinite dilution when perfect solution is at the standard state.

The molar excess Gibbs energy for a system of $n$ chemical species can be calculated by Equation (30),

$$G_E = \sum_{i=1}^{n} x_i RT \, \ln(\gamma_i),$$ (30)

Using Equation (30) and considering Equations (18) and (28), the equation for the molar excess Gibbs energy of the system is obtained,

$$\frac{G_E}{RT} = \sum_{i=1}^{l} x_i \sum_{f=1}^{3} \ln \gamma_{i,f}^{ps} + \sum_{i=l+1}^{r} x_i \sum_{f=1}^{3} \ln(\ln \gamma_i^{ps} - \ln \gamma_i^{\infty}) + \sum_{i=r+1}^{n} x_i \sum_{f=1}^{2} (\ln \gamma_{i,f}^{ps} - \ln \gamma_{i,f}^{\infty}) + \sum_{i=r+1}^{n} x_i \ln \gamma_{i,D-H}^{ids},$$ (31)

where the subscript $f$ indicates the type of interaction ($f = 1$ means combinatorial; $f = 2$ means residual and $f = 3$ means Debye–Huckel).

The activity coefficient for species $i$ is obtained by Equation (32),

$$\ln \gamma_i = \frac{\partial (nG_E / RT)}{\partial n_i} \bigg|_{P,T,nj \neq i},$$ (32)

Therefore, the activity coefficients in Equation (31) can be expressed as,

$$\ln \gamma_{i,C}^{ps} = \ln(\frac{\phi_i}{x_i}) + \frac{z}{2}q_i \ln(\frac{\theta_i}{\phi_i}) + l_i - \frac{\phi_i}{x_i}(\sum_{j=1}^{n} \ln x_j l_j),$$ (33)

$$l_i = \frac{z}{2}(r_i - q_i) - (r_i - 1),$$ (34)

$$\ln \gamma_{i,R}^{ps} = q_i[1 - \ln(\sum_{j=1}^{n} \theta_j \tau_{ji}) - \sum_{j=1}^{n} \frac{\theta_j \tau_{ij}}{\sum\limits_{k=1}^{n} \theta_k \tau_{kj}}],$$ (35)

$$\ln \gamma_{i,DH}^{ps} = \frac{2AM_s}{b^3}[(1 + bI^{\frac{1}{2}}) - (1 + bI^{\frac{1}{2}})^{-1} - 2\ln(1 + bI^{\frac{1}{2}})],$$ (36)

$$\ln \gamma_{i,DH}^{ids} = -Az_i^2 \frac{I^{\frac{1}{2}}}{1 + bI^{\frac{1}{2}}},$$ (37)

$$\ln \gamma_{i,C}^{\infty} = \ln(\frac{r_i}{r_s}) + \frac{z}{2}q_i \ln(\frac{q_i r_s}{q_s r_i}) + l_i - \frac{r_i}{r_s}l_i,$$ (38)

$$\ln \gamma_{i,R}^{\infty} = q_i[1 - \ln(\tau_{si}) - \tau_{is}],$$ (39)

$$\ln \gamma_{i,DH}^{\infty} = 0$$ (40)

For the solution of NaCl, the molar fractions of three dissociated species ($x_w$, $x_{Na^+}$, $x_{Cl^-}$) must be considered, and the water actively can be determined by Equation (41)

$$\ln a_w = \ln(\frac{\phi_w}{x_w}) + \frac{z}{2}q_w \ln(\frac{\theta_w}{\phi_w}) + [\frac{z}{2}(r_w - q_w) - (r_w - 1)] - \frac{\phi_w}{x_w}(\sum_{i=1}^{n} x_i l_i) + q_w[1 - \ln(\sum_{i=1}^{n} \theta_i \tau_{iw}) - \sum_{i=1}^{n} \frac{\theta_i \tau_{wi}}{\sum\limits_{j=1}^{n} \theta_j \tau_{ji}})]$$
$$+ \frac{2AM_w}{b^3}[(1 + bI^{\frac{1}{2}}) - (1 + bI^{\frac{1}{2}})^{-1} - 2\ln(1 + bI^{\frac{1}{2}})] + \ln x_w$$ (41)

When the water is considered as the solvent, the ionic strength can be calculated by

$$I = \frac{1}{2 \cdot x_w M_w}\sum_{i=1}^{n} x_i z_i^2,$$ (42)

### 4.3. Calculation Results

As shown in the theoretical analysis (Sections 4.1 and 4.2), the freezing point depression can be calculated by the water activity Equation (17). Based on the published literature, the parameters used in the extended UNIQUAC model were obtained, as shown in Tables 2–4. The computational procedure was present in Figure 8.

**Table 2.** Parameters *q* and *r* of typical species in UNIQUAC [16].

| Species | *q* | *r* |
|---|---|---|
| $H_2O$ | 1.400 | 0.9200 |
| $Na^+$ | 1.199 | 1.4034 |
| $K^+$ | 2.4306 | 2.2304 |
| $Mg^{2+}$ | 2.540 | 5.410 |
| $Cl^-$ | 10.197 | 10.386 |

**Table 3.** Interaction parameters $u_{ji}^0$ of UNIQUAC model [16,23].

| i/j | H₂O | Na⁺ | K⁺ | Cl⁻ |
|-----|-----|-----|-----|-----|
| H₂O | 0 | 733.286 | 535.023 | 1523.39 |
| Na⁺ |  | 0 | −46.194 | 1443.23 |
| K⁺ |  |  | 0 | 1465.18 |
| Cl⁻ |  |  |  | 2214.81 |

**Table 4.** Interaction parameters $u_{ji}^t$ of UNIQUAC model [16,23].

| i/j | H₂O | Na⁺ | K⁺ | Cl⁻ |
|-----|-----|-----|-----|-----|
| H₂O | 0 | 0.4872 | 0.9936 | 14.631 |
| Na⁺ |  | 0 | 0.1190 | 15.635 |
| K⁺ |  |  | 0 | 15.329 |
| Cl⁻ |  |  |  | 8.3194 |

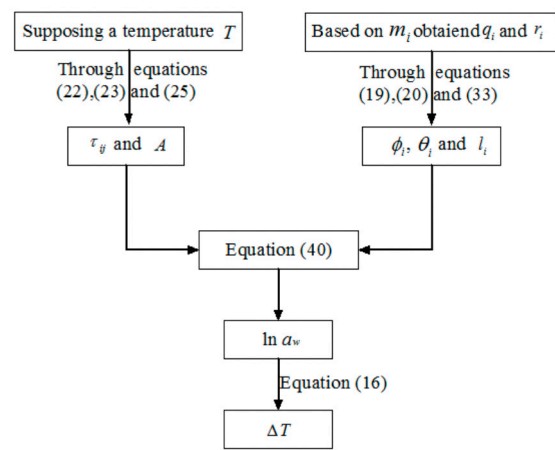

**Figure 8.** Schematic diagram of freezing point calculation process.

In Sections 4.1 and 4.2, the prediction formula of freezing point was presented, and the decreasing freezing point can be calculated by Equations (17) and (41). Here, to verify the correctness of this model, the NaCl was treated as the solute. The comparisons between the calculated values and the test data are presented in Figure 9. As shown, the calculated values are very close to the experimental results. The results show that the predicted freezing point depression are very good in the whole concentration range. This indicates that the presented equations are reasonable.

Due to the limitation of the test conditions, the salt concentration is distributed in a small range. Even so, from the results of Figure 9a,b, it can be concluded that the decreasing freezing point is not influenced by the soil type but is sensitive to the salt concentration. To supplement this conclusion, the experimental freezing point of the NaCl-H₂O solution is cited and presented in Figure 9c. As shown in Figure 9c, the predicted values agreed well with the experimental data, especially in the small concentration range.

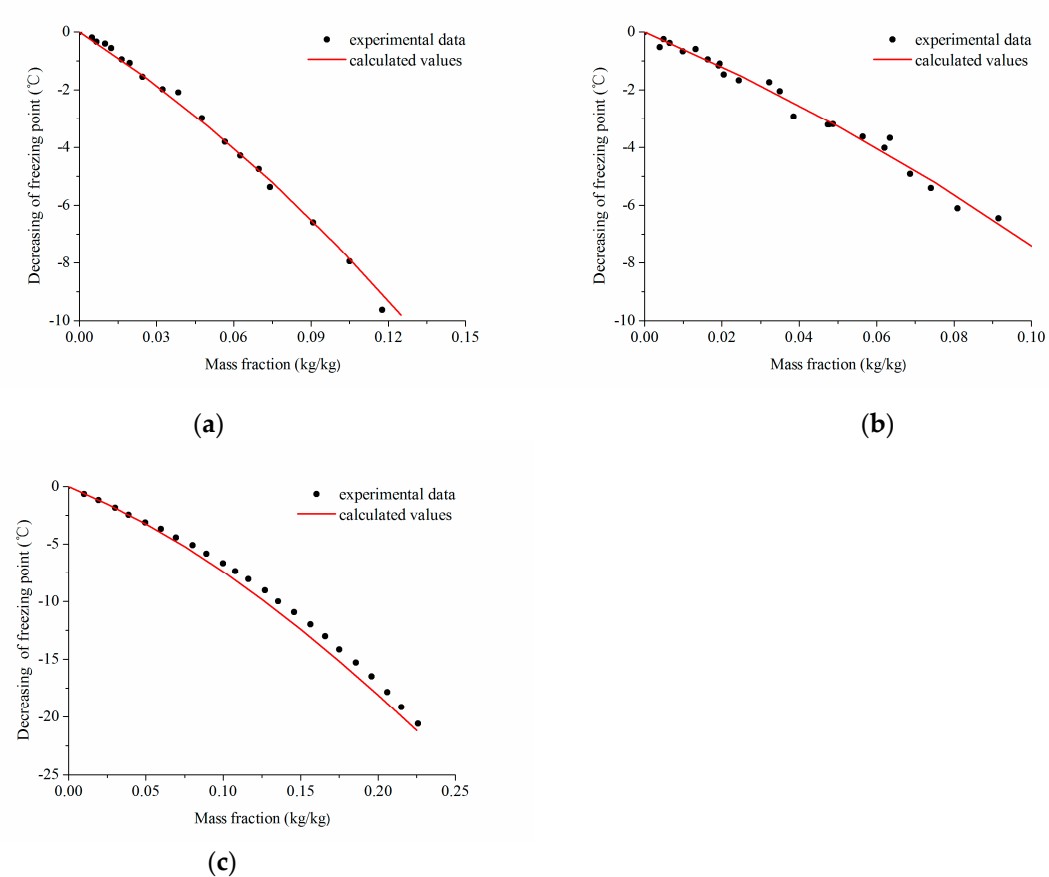

**Figure 9.** Comparison between the experimental data and the calculation value (**a**) Experimental data from this paper, (**b**) Experimental data from Bing and Ma (2011) [13], (**c**) Experimental data from Haynes et al. (2015) [18].

## 5. Discussion

In this study, the freezing point depression caused by the solute can be calculated by Equation (17). Once the freezing point $T_{F0}$ of desaline soil is known, then the freezing point $T_F$ of saline soil can be calculated by Equations (15) and (17). As shown in Equation (17), the water activity is a key parameter to calculate the freezing point. The water activity was determined by the UNIQUAC model. From the existing literature, it is well known that the water activity can be calculated by the Pitzer model, the UNIQUAC model and other models. However, the parameters in the Pitzer model related to the temperature and obtained the value at the 298.15 K [24]. As a result, the calculation results will produce a large difference with the experimental results. Based on the origin UNIQUAC model, the researchers have developed this model and these parameters are available for many single salts. Consequently, the UNIQUAC model was adopted in this paper to calculate the water activity and the freezing point.

To verify the validation of the presented model, Figure 9 presented the comparison results between the experimental results and the calculated values. From the comparison results, it can be known that the calculated values are consistent with the experimental results at the lower concentration (Figure 9). However, under the high concentration, the deviations are slightly larger. The whole calculation values are slightly larger than the experimental value Figure 9c. Table 5 shows the comparison results of freezing point between the calculated values and the tested values.

**Table 5.** Comparison of calculated and measured freezing point (for NaCl).

| Sample Size | Concentration (kg/kg) | Relative Error (%) | Data Sources |
|---|---|---|---|
| 18 | 0.005~0.118 | −1.89~2.15 | Test result in this paper |
| 24 | 0.004~0.092 | −1.57~2.98 | Bing and Ma, 2011 |
| 24 | 0.002~0.230 | 0.23~4.85 | Haynes et al., 2015 |

As shown in Table 5, the deviation between the measured value and the calculated value increases with the increase of salt concentration. However, the deviation is less than 5%. In our opinion, the deviation can be attributed to the following aspects:

(1) Deviation from the assuming temperature. As shown in the computation flow (Figure 8), an assumed temperature should be given out to calculate the water activity. If the assumed temperature is not suitable, then the water activity may be incorrect. To reflect the influence of the assumed temperature on the water activity, Figure 10 shows the variation of water activity with different assumed temperature and salt concentrations. The difference of active water under different temperatures becomes larger with the increase of salt concentration. This indicates that the assuming temperature can bring some errors, especially at a high concentration.

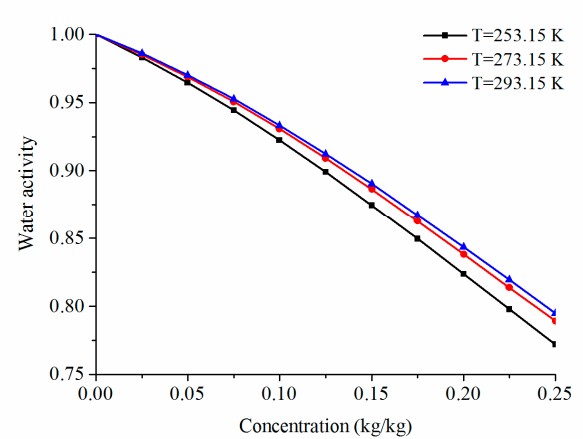

**Figure 10.** Influence of temperature and concentration on water activity.

(2) Deviation from the NaCl solute. The NaCl solute can affect the calculation result of the thermodynamic model by affecting the hydration process and ionic radius. For the high concentration solution, influenced by the solubility, the hydration is not complete in the laboratory test. However, this is ignored in the calculation model. As a result, there are some differences between the measured value and the calculated value, especially at a higher salt concentration.

As shown in the theoretical analysis, the excess Gibbs energy consists of three parts [18]: (1) combinatorial term accounts for molecular size and shape differences, (2) the residual term accounts for the molecular interactions and (3) the Debye–Hückel term accounts for the long-range electrostatic interactions. Both test results and theoretical analyses indicate that salt content can decrease the freezing point. Now, the question is, which part has a decisive impact on the decreasing of freezing point? To answer this question, Figure 11 provides the variation of the three different terms at different concentrations. As seen, the Debye–Hückel term increases with the increases of salt concentration. However, both the combinatorial term and the residual term decrease with the increase of concentration. The result indicates that the salt concentration has a large influence on the residual term. When the concentration is larger than 0.10 kg/kg, the residual term rapidly decreases with the increase of concentration. For the NaCl solution, the mole fraction of water decreases with the increase of concentration. As a result, the residual term decreases with the increase of salt concentration. Moreover, it can be found that compared with the other terms, the residual term has a clear reduction with the

increase of salt concentration. This indicates that salt decreases the molecular interactions and results in the decrease of water activity.

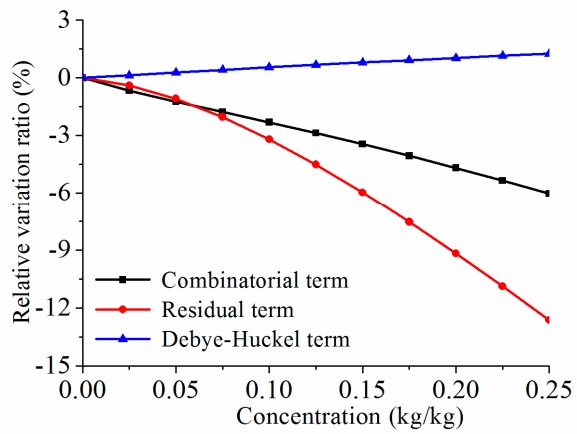

**Figure 11.** Change ratio of three terms in Equation (18).

Based on the thermodynamic model and water activity, the freezing point depression of soil caused by the solute was derived. However, the present model is not very suitable for solutions with a very high concentration. In addition, the presented model cannot calculate the freezing point of desalinated soil. Apart from these defects, this model can provide a good approximation for predicting the freezing point depression. In later work, we will modify this predicted model and overcome these shortages.

## 6. Conclusions

In this study, the freezing point of saline soil was investigated by the laboratory experiment and theoretical analyses. From the analysis, the following conclusions can be drawn:

(1)　There exists a critical water content, no matter the salt contents. When the initial water content is lower than the critical water content, the freezing point increases with the increase of the water content. When the initial water content is larger than the critical water content, the increase of initial water content has little influence on the freezing point.

(2)　The freezing point is related to the energy status of liquid water in saline soils. A thermodynamic model of excess Gibbs energy was proposed for predicting freezing point of saline soil. Compared to the experimental result, a satisfactory accuracy was observed for the systems studied and the validity of the presented model was verified.

(3)　At the same conditions, the addition of salt reduces the total potential of soil water and decreases the molecular interactions. Consequently, the increase of salt content decreases the water activity. As a result, the freezing point decreases. Moreover, the freezing point depression of saline soil is mainly caused by the decrease of molecular interaction.

**Author Contributions:** Conceptualization, F.M. and L.C.; Formal analysis, F.M.; Methodology, F.M. and L.C.; Writing—original draft, F.M.; Writing—review & editing, L.C., D.L. and C.D. All authors have read and agreed to the published version of the manuscript.

**Funding:** This work is supported by the Funding of the State Key Laboratory Frozen Soil Engineering (grant numbers SKLFSE-ZT-17) and the National Natural Science Foundation of China (grant numbers 41701060).

**Conflicts of Interest:** The authors declare no conflict of interest.

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
