# Peer review of "Investigation into Freezing Point Depression in Soil Caused by NaCl Solution"

_water, doi:10.3390/w12082232_

Round 1
Reviewer 1 Report
Here are a few observations:
- this is a well organized and interesting paper supported by experimental data with clear analysis and the application of the UNIQUAC model. The effectiveness of the model has been considered, including an accuracy figure.
- In places the overall clarity of written English could be improved, there are some examples below but this is not a comprehensive list of corrections.
- line 16: "serials" should be "series", "Besides" not required
- line 24: "decrease" should be "decreased"
- line 25: "solution lowers" should be "solution which lowers"
- line 26: "with other" should be "with the other"
- line 29: "how the salt affect" should be "how salt concentration affects"
- line 30: "provide" should be "provides"
- line 18: UNIQUAC (Universal Chemical Activity Coefficient) the expanded wording does not match the abbreviation, 'universal quasichemical activity coefficient'?
- line 46: "owing to" should this be "knowing"? Or, should "owing to the freezing temperature is the key point to judge " be "owing to the freezing temperature being the key point to judge"
- line 47 "down" not required
- line 54 "is" not required
- lines 102 & 103: is it possible to include manufacturer details, i.e. part numbers etc., of: the temperature sensor, data acquisition and cold bath controller system?
- Figure 4a: could the saturated water content figure be added in the caption? I presume this could be 28.7% referred to in line 110?
- Equation 5: are both subscripts correct?
- Figure 8 labeling of graphs, a, b & c is missing
- Line 276: "related" should this be "relation", or, "model in related" should this be "model related"?
- Line 280: "adopt" should be "adopted"
Author Response
Responses to Reviewer 1
Comments and Suggestions for Authors
Here are a few observations:
Comment 1: This is a well-organized and interesting paper supported by experimental data with clear analysis and the application of the UNIQUAC model. The effectiveness of the model has been considered, including an accuracy figure.
Response:
Dear Reviewer, thanks very much for your approval.
Comment 2: In places the overall clarity of written English could be improved, there are some examples below but this is not a comprehensive list of corrections.
Response: Thanks so much for your valuable comments, which are significant for us to promote our manuscript in writing. Meanwhile, we are so sorry for the inconvenient for you to review this manuscript. So, we have revised the manuscript by your precious comments to our best efforts. The responses are listed point by point
Comment 3: line 16: "serials" should be "series", "Besides" not required.
Response: Thanks very much for your precious comment. We are so sorry for making such a mistake. After your friendly reminder, the word "series" has been corrected in the revised manuscript.
In addition, the word "Besides" has been deleted.
Comment 4: line 24: "decrease" should be "decreased".
Response: Thanks very much for your precious comment. We are so sorry for making such a mistake. After your friendly reminder, we used "decreased" instead of "decrease" in the revised manuscript.
Comment 5: line 25: "solution lowers" should be "solution which lowers".
Response: Thanks very much for your precious comment. We are so sorry for making such a mistake. After your friendly reminder, the words "which" has been added in revised manuscript.
Comment 6: line 26: "with other" should be "with the other".
Response: Thanks very much for your precious comment. We are so sorry for making such a mistake. After your friendly reminder, the words "the" has been added in revised manuscript.
Comment 7: line 29: "how the salt affect" should be "how salt concentration affects".
Response: Thanks very much for your precious comment. We are so sorry for making such a mistake. After your friendly reminder, we used "how salt concentration affects" instead of "how the salt affect" in the revised manuscript.
Comment 8: line 30: "provide" should be "provides".
Response: Thanks very much for your precious comment. We are so sorry for making such a mistake. After your friendly reminder, we used "provides" instead of "provide" in the revised manuscript.
Comment 9: line 18: UNIQUAC (Universal Chemical Activity Coefficient) the expanded wording does not match the abbreviation, 'universal quasichemical activity coefficient'?
Response: Thanks very much for your precious comment. Yes, the expanded wording is incorrect. We are so sorry for making such a mistake. The right form is UNIQUAC (universal quasi-chemical). This has been corrected in the revised manuscript.
Comment 10: line 46: "owing to" should this be "knowing"? Or, should "owing to the freezing temperature is the key point to judge "be"owing to the freezing temperature being the key point to judge".
Response: Thanks very much for your precious comment. This sentence has been revised as "owing to the freezing point being the key point to judge whether the soil is frozen or not".
Comment 11: line 47 "down" not required.
Response: Thanks very much for your precious comment. We are so sorry for making such a mistake. Yes, the word "down" is not required, and it has been deleted in the revised manuscript.
Comment 12: line 54 "is" not required.
Response: Thanks very much for your precious comment. We are so sorry for making such a mistake. Yes, the word "is" is not required. After revise, this sentence has been changed to "That's why makes the measurement of freezing point becomes difficult" (line 50-51).
Comment 13: Lines 102 & 103: is it possible to include manufacturer details, i.e. part numbers etc., of: the temperature sensor, data acquisition and cold bath controller system?
Response: Thanks very much for your kindly reminder. Generally, the freezing point of saline soil was measured by a self-made device (made by the State Key Laboratory of Frozen Soil Engineering).
The temperature sensor is made by the State Key Laboratory of Frozen Soil Engineering.
The data acquisition used DT80 which was made by Thermo Fisher Scientific Company, Australia.
The type of the cold bath is FP55-SL made by Julabo Company in America.
Comment 14: Figure 4a: could the saturated water content figure be added in the caption? I presume this could be 28.7% referred to in line 110?
Response: Thanks very much for your precious comment. The term "saturated water content" is not suitable to describe this character. Because the saturated water content is changing with dry density.
In this study, the dry density is 1.0-1.5g/cm3. Therefore, the saturated water content is not a constant.
Here, we use the word "critical water content" to describe this character. The line of critical water content has been added in Fig.5a.
(a) different water contents
Figure 5. Influence of water content and Nacl content on the freeing temperature.
Comment 15: Equation 5: are both subscripts correct?
Response: Thanks very much for your precious comment. The subscripts are incorrect. The right form as follows.
|
, |
(5) |
Comment 16: Figure 8 labeling of graphs, a, b & c is missing.
Response: Thanks very much for your precious comment. The labeling of graphs, a, b & c in Fig.9 have been added in line 264 and 265
Comment 17: Line 276: "related" should this be "relation", or, "model in related" should this be "model related"?
Response: Thanks very much for your precious comment. We are so sorry for making such a mistake. The word "in" redundant. It has been deleted in the revised manuscript (line 281.
Comment 18: Line 280: "adopt" should be "adopted".
Response: Thanks very much for your precious comment. We are so sorry for making such a mistake. After your friendly reminder, we used "adopted" instead of "adopt" in the revised manuscript (line 285.

Reviewer 2 Report
The matter of the manuscript ‘Investigation into freezing temperature depression in soil caused by NaCl solution’ is noteworthy and fits into the scope of the Water journal. The results presented are appropriate and interesting, which, in general, deserves publication. The experimental procedure and analysis are mostly correct. However, the current version of the manuscript requires revisions and additions.
Thus, I am frustrated with the poor description of Materials and Methods. Soil characteristics are of crucial importance when considering not only the freezing patterns but also the usability of the substrate. What are the soil types? From which horizons were the samples taken? What was known about particle distribution, organic carbon content, and soil density? Please provide the full description of study objects as you have investigated not an abstract bucket of organo-mineral materials but rather a complex natural system.
I honestly hope you will find my suggestions supportive.
Kind regards,
Reviewer
Author Response
Responses to Editors’ comments
Dear editor:
Thank you for giving us an opportunity to revise the manuscript "Investigation into freezing point depression in soil caused by NaCl solution" [Ms. Ref. water-861714].
We highly appreciate your comments and review’s suggestions, which are very helpful for improving our paper. Based on these comments and suggestions, we have made careful modifications on the original manuscript. Appended to this letter is our point-to-point response to the comments. These comments were reproduced and expressed in the manuscript.
We have discussed the reviewer’s comments carefully and have made revisions which appear in red in the revised paper. The manuscript has been revised to the best with our knowledge according to the reviewers’ suggestions as follows for your evaluation. We are grateful for your attention to our manuscript. Once again, thanks very much for your arduous work and instructive suggestions to our manuscript processing.
Thank you and best regards.
Yours sincerely
Lei Chen (on the behalf of all authors)
Jul.16, 2020
Responses to Reviewer 2
Comments and Suggestions for Authors
Comment 1: The matter of the manuscript ‘Investigation into freezing temperature depression in soil caused by NaCl solution’ is noteworthy and fits into the scope of the Water journal. The results presented are appropriate and interesting, which, in general, deserves publication. The experimental procedure and analysis are mostly correct.
Dear Reviewer,
Firstly, thanks very much for reviewing our manuscript. Receiving such positive feedback from you motivates us to further improve our works.
Secondly, thanks very much for your precious comments and guidance. They are so important for us to revise our manuscript. In the meantime, we are so sorry for any inconvenient for you to review this manuscript. So, we have made a serious revise to the manuscript especially in the section of Materials and Methods. We sincerely hope you will be satisfied with our revisions.
Comment 2: However, the current version of the manuscript requires revisions and additions. Thus, I am frustrated with the poor description of Materials and Methods. Soil characteristics are of crucial importance when considering not only the freezing patterns but also the usability of the substrate. What are the soil types? From which horizons were the samples taken? What was known about particle distribution, organic carbon content, and soil density? Please provide the full description of study objects as you have investigated not an abstract bucket of organo-mineral materials but rather a complex natural system.
Response: Thanks very much for your kindly reminder.
Yes, the description of the Materials and Methods is poor in the previous version.
Based on your comment, we add the sections of Experimental material (2.1) and Sample preparation (2.2). The section of Experimental scheme (2.3) has been further refined.
These modified parts are listed in line 84-114, as follows:
2.1. Experimental material
The soil used in this study was collected from the Loess Plateau in Lanzhou (36°03′ north latitude, 103°40′ east longitude). To investigate the effect of salt content on the freezing point, all the soil was washed by the distilled water to eliminate the salt. The desalinated soil was dried at 105 ℃ for 24 hours and sieved through a 2-mm sieve. The soil particles with sizes of less than 2 mm were used as the experimental specimens. The particle size distribution curve was shown in Fig. 1. This silty clay has a plastic limit of 15.4%, a liquid limit of 28.7% and a specific gravity of 2.62 g/cm3.
Fig. 1 Particle size distribution curve of the tested soil.
2.2. Sample preparation
Based on the results of field survey, dry density, water content and salt concentration of the specimen were determined as 1.0~1.5 g/cm3, 5~50% and 0~4%, respectively. According to the test scheme, the specimen was mixed with different mass of water, salt (NaCl) and soil (without water) at room temperature and sealed for 24 hours to make the sample reach a uniform state.
2.3. Experimental scheme
The freezing point of saline soil was measured by a self-made device (made by the State Key Laboratory of Frozen Soil Engineering). The schematic of the test apparatus for freezing point measurement was presented in Fig. 2. As seen, the test apparatus mainly consists of sample box, sample cover, temperature sensor (with an accuracy of ± 0.02 ℃) and data acquisition (DT 80). The cold bath (with an accuracy of ±0.1 ℃) was used to provide a sub-freezing temperature. The sample box was 5 cm in high and with an inner diameter of 3.5 cm. The sample box and the cover were made by cooper. In order to inset the temperature sensor, there was a hole in the sample cover. The gap between the temperature sensor and the sample cover was filled with rubber.
Figure 2. Test apparatus for freezing point measurement
After the specimen was put into the sample box, the sample cover was covered on the sample box. Then, temperature sensor was inserted into the sample box. The gap between the temperature sensor and the sample cover was filled with rubber. After that, the sample box was placed in the cold bath at 15 ℃ for twelve hours. Finally, the temperature of the cold bath was set as -15 ℃. The temperature data was recorded automatically by the data acquisition system with an interval of 10 seconds.

Round 2
Reviewer 2 Report
I can now suggest that the manuscript is accepted. However, it would be good to see the clear definition of soil type according to either WRB of FAO classification.
Author Response
Responses to Reviewer 2
Comments and Suggestions for Authors
Comment: I can now suggest that the manuscript is accepted. However, it would be good to see the clear definition of soil type according to either WRB of FAO classification.
Response:
Dear Reviewer,
Thank you very much for valuable comments. Yes, it would be better to see the clear definition of soil type in the manuscript. Thus, we classified the soil based on the soil classification of WRB.
Action:
“According to the WRB-based soil classification [15], the soil can be classified into the aridic.”( L91)
